# Phosphorus Supplementation Enhances Growth and Antioxidant Defense Against Cadmium Stress in Cotton

**DOI:** 10.3390/antiox14060686

**Published:** 2025-06-05

**Authors:** Asif Iqbal, Huiping Gui, Cangsong Zheng, Xiangru Wang, Xiling Zhang, Meizhen Song, Xiaoyan Ma

**Affiliations:** 1Institute of Cotton Research of Chinese Academy of Agricultural Sciences/Zhengzhou Research Base, State Key Laboratory of Cotton Bio-Breeding and Integrated Utilization, School of Agricultural Sciences, Zhengzhou University, Anyang 455000, China; asif173aup@gmail.com (A.I.); huiping.828@163.com (H.G.); zhengcangsong@163.com (C.Z.); wxr_z4317@163.com (X.W.); hainan1571@163.com (X.Z.); 2Xinjiang Key Laboratory of Crop Gene Editing and Germplasm Innovation, Institute of Western Agricultural of CAAS, Changji 831100, China; 3Department of Agriculture, Hazara University, Khyber Pakhtunkhwa, Mansehra 21120, Pakistan

**Keywords:** cotton, phosphorus, cadmium stress, reactive oxygen species, antioxidant system, osmotic adjustment

## Abstract

Cadmium (Cd) contamination in agricultural soils is increasing due to anthropogenic activities, posing a significant threat to plant growth and productivity. Phosphorus (P) has been suggested as a potential mitigator of Cd toxicity, yet the role of cotton genotypes with contrasting low-P tolerance in contaminated soils remains largely unexplored. A hydroponic experiment was conducted to assess the effects of Cd stress (5 μM) on Jimian169 (strong-low-P tolerant) and DES926 (weak-low-P tolerant) cotton genotypes under low-P (0.01 mM KH_2_PO_4_) and normal P (1 mM KH_2_PO_4_) conditions. The results revealed that Cd stress, especially under low-P, significantly reduced plant growth, dry matter, photosynthetic rate, and P use efficiency (PUE), while increasing oxidative damage through increased malonaldehyde levels and reactive oxygen species accumulation. These adverse impacts were very much evident in DES926 compared to Jimian169. In contrast, Jimian169 demonstrated greater resilience to Cd stress by mitigating oxidative damage through enhanced antioxidant enzyme activity, improved photosynthetic performance, and increased accumulation of osmoprotectants. These findings indicate that Jimian169 can better withstand Cd toxicity by enhancing photosynthesis, antioxidant defense mechanisms, and osmotic adjustment. This makes them a promising candidate for cultivation in Cd-contaminated, P-deficient soils.

## 1. Introduction

Cadmium (Cd) is recognized as a highly toxic heavy metal with significant environmental and agricultural implications due to its high solubility in water and rapid uptake by plants [1]. It is one of the most readily translocated heavy metals within the plant tissues, leading to severe physiological disruptions [2]. Cd accumulation inhibits cell division in meristematic tissues [3], disrupts photosynthesis, and interferes with key metabolic processes, leading to excessive production of reactive oxygen species (ROS) and resulting in oxidative stress [4]. These detrimental effects necessitate a robust defense system to mitigate oxidative damage and maintain cellular homeostasis.

In order to protect their photosynthetic structures from the negative impact of ROS and photoinhibition, plants have developed a variety of approaches. One of these approaches is the adjustment of the light-harvesting antenna size, which diminishes light absorption when CO_2_ assimilation is insufficient [5]. Moreover, both the photochemical and the non-photochemical quenching mechanisms efficiently dissipate excess excitation energy from the photosynthetic process, thereby preventing the accumulation of ROS [6]. The detoxification of ROS and maintenance of cellular redox balance are facilitated by enzymatic antioxidants, such as superoxide dismutase (SOD), peroxidase (POD), catalase (CAT), as well as non-enzymatic antioxidant systems including glutathione, ascorbate, and carotenoids [7]. SOD catalyzes the dismutation of superoxide radicals (O_2_^−^), while POD and CAT scavenge hydrogen peroxide (H_2_O_2_) in mitochondria, collectively mitigating oxidative stress [8].

Cd, a redox-inactive heavy metal induces the generation of ROS, including O_2_^−^, H_2_O_2_ and hydroxyl radical [9]. In chloroplast, O_2_^−^ is primarily produced due to electron leakage at photosystem II (PSII), either at the acceptor side or through incomplete water oxidation on the donor side [10]. Cd disrupts PSII functions by replacing calcium in the oxygen-evolving complex and impairing electron transport between the primary quinone acceptor and secondary quinone acceptor [11]. Although O_2_^−^ itself causes minimal damage, it acts as a precursor for more reactive species, such as singlet oxygen (1O_2_) and OH, which can lead to lipid peroxidation and protein disfunction. To counteract this, SOD plays a critical role in ROS scavenging by converting O_2_^−^ into less harmful molecules like H_2_O_2_, making it a key player in ROS scavenging [12].

Phosphorus (P) is a key major nutrient that regulates several metabolic processes, including leaf photosynthesis, carbohydrate metabolism, and energy transfer [13]. Besides its agronomic significance, P application in soil is gaining attention for its role in altering heavy metals bioavailability [14]. P enhances soil pH, leading to the immobilization of heavy metals such as Cd, thus mitigating their toxic effects on plants [15]. Additionally, P plays a vital role in reinforcing antioxidant defenses during abiotic stress conditions [16]. Previously, it was reported that P supplementation alleviates the toxicity of Cd in wheat [17], copper toxicity in soybean [18], and aluminum in eucalyptus [19] and Camellia oleifera [20]. Under Cd stress, P enhances chlorophyll contents and activates antioxidant enzymes, thereby improving tolerance in wheat [17]. Likewise, P mitigates toxicity of manganese (Mn) in soybean [21], potato [22], and pasture species such as perennial ryegrass and white clover [23]. High P doses also enhance photosynthetic efficiency and reduce ROS production under Mn toxicity [24], suggesting its broader role in mitigating oxidative stress.

Despite these benefits, P supplementation can enhance heavy metals accumulation like Cd in plants [25]. Studies indicate that P enhances Cd retention in cell walls by increasing cell wall pectin content in mangrove roots under Cd exposure [26]. P also facilitates Cd sequestration by forming Cd-phosphate complexes, that are subsequently deposited in vacuoles and root surfaces [27]. Additionally, P plays a critical role in antioxidant defenses by stimulating the synthesis of radical scavengers such as proline, and various antioxidants both enzymatic and non-enzymatic [25]. P also contributes to carbohydrate synthesis, which supports antioxidative processes [28]. Research on *Myracrodruom urundeuva* reveals that P boosts glutathione biosynthesis, thereby enhancing enzymatic antioxidant activity [29]. In poplar, P deficiency inhibits the ascorbate-glutathione cycle, leading to elevated oxidative stress [25]. Moreover, P reduces malonaldehyde levels and alleviates oxidative damage in Cd-exposed mangrove seedlings by promoting antioxidants and phytochelatin production [30].

Based on the above discussion, this study was designed to investigate whether exogenous P supplementation enhances Cd tolerance in cotton, a globally important cash crop. Specifically, the study investigates how P influences growth, photosynthesis, phosphorus use efficiency, and antioxidant enzyme activities under Cd stress. To explore this, a hydroponic experiment was conducted using two Cd concentrations (0, and 5 μM) with and without 1 mM of P supplementation. We hypothesized that P application, along with low-P tolerant cotton genotypes, can enhance Cd stress tolerance by improving plant morphology and physiology such as photosynthesis, osmotic regulation, and antioxidant system. Consequently, the current study was designed to assess the responses of cotton genotypes and their capacity to mitigate Cd stress with P supplementation.

## 2. Materials and Methods

### 2.1. Plant Materials and Growth Conditions

A hydroponic experiment was carried out to assess the impact of Cd stress on Jimian169 (strong-low-P tolerant) and DES926 (weak-low-P tolerant) under low and normal P [31]. The seeds for these genotypes were obtained from the Cotton Research Institute of Chinese Academy of Agricultural Sciences, with official authorization. To ensure uniform germination, healthy seeds were initially sown in sterilized sand and kept under controlled conditions for one week. Once germinated, seedlings with uniform growth were transplanted into 7 L plastic containers filled with a nutrient solution. The plants were maintained under a regulated environment with a 16 h light and 8 h dark cycle, a temperature of 28 °C, and a relative humidity of 60%. Initially, a half-strength solution for the first week was applied, which was then replaced with a full-strength Hoagland solution for the remainder of the experiment [32,33]. Upon reaching the three-leaf stage, the seedlings were categorized into four treatment groups:(i)LP + Cd (0.01 mM KH_2_PO_4_ + 5 μM CdCl_2_)(ii)LP + CK (0.01 mM KH_2_PO_4_ + 0 μM CdCl_2_)(iii)NP + Cd (1 mM KH_2_PO_4_ + 5 μM CdCl_2_)(iv)NP + CK (1 mM KH_2_PO_4_ + 0 μM CdCl_2_)

To ensure proper aeration, an electric pump was used, and nutrient solutions were refreshed weekly. Additionally, to eliminate position bias, the placement of the containers was rotated during solution renewal. After two weeks of treatment, noticeable morphological differences emerged between the genotypes which were then harvested to study various morphological and physiological traits.

### 2.2. Growth Measurements and Root Analysis

Randomly, six plants were selected from each treatment to assess shoot length using the scale [34]. At the end, plants were harvested, and the roots and shoots were separately dried at 105 and 80 °C for 1 and 48 h, respectively. The root, shoot, and total biomass were then measured using an electric balance. To evaluate root morphology, root samples were scanned through EPSON and the analysis was performed in WinRhizo [32].

### 2.3. Leaf Physiological Assessments

Leaf gaseous exchange parameters that included net photosynthetic rate, transpiration rate, stomatal conductance, and intercellular CO_2_ concentration were recorded from the third fully expanded leaf by using Li-Cor 6800 from morning 9–11 AM [35]. Similarly, leaf chlorophyll and carotenoid contents were measured using the protocol mentioned in our previous study [36].

### 2.4. Phosphorus Concentration and Use Efficiency

The root and shoot P concentration was quantified by the Kjeldahl method [37]. For this analysis, amounts of 0.2 g of ground samples were digested in a mixture of sulphuric acid and hydrogen peroxide and the resulting P concentrations were determined using Auto Analyzer III. The PUE traits were then calculated using the formulas highlighted in the previous study [38].

### 2.5. Malonaldehyde and Reactive Oxygen Species Determination

Root and shoot malondialdehyde concentrations were evaluated by thiobarbituric acid (TBA) reactivity [39]. A 0.2 g sample was extracted in 2 mL (0.25% TBA) dissolved in 10% trichloroacetic acid (TCA) and incubated at 95 °C for 30 min. The homogenate was centrifuged at 10,000× *g* for 10 min, and the data were calculated at 532 nm [40].

Hydrogen peroxide (H_2_O_2_) concentration was determined using the standard protocol [41]. Around 0.1 g of tissue samples were each mixed in 5 mL of 0.1% TCA and centrifuged for 15 min at 12,000× *g*. The amount of H_2_O_2_ was then measured spectrophotometrically at 390 nm following the standard protocol [31].

Superoxide (O_2_^−^) concentration was measured using a protocol adapted from the previous study [41]. Briefly, a 0.2 g sample was mixed in 1 mL of 65 mM phosphate buffer (pH 7.8) and then centrifuged at 10,000× *g* for 10 min. The absorbance of the final solution was calculated at 450 nm using spectrophotometer.

### 2.6. Antioxidant Enzyme Activity Assays

To measure enzymatic activity, 0.5 g of root and shoot tissues were ground using liquid nitrogen and mixed in 10 mL of 50 mM sodium phosphate buffer containing 1% polyvinylpyrrodine, 0.2 mm per L EDTA, and 19 mmol per L magnesium chloride. The mixture was centrifuged and kept at 4 °C for subsequent enzyme analysis.

Peroxidase activity was assessed using a method adopted from our previous study [42]. Superoxide dismutase (SOD) activity was measured using a photochemical nitroblue tetrazolium (NBT) assay and was monitored at 560 nm. The CAT activity was determined by observing the reduction at 240 nm due to H_2_O_2_ degradation as mentioned in our earlier study [43].

### 2.7. Osmoprotectants Quantification

Free amino acids were quantified using the protocol of previously published work with some modifications [44]. The extraction buffer was composed of acetic acid/sodium acetate (pH 5.4), and the final data were collected at 580 nm via a spectrophotometer [42].

Total soluble protein content was measured following the method of Bradford [45] with bovine serum albumin as the standard and the absorbance was recorded at 595 nm [46].

Total soluble sugars were analyzed based on a modified protocol adopted from a previous study [47]. A 0.5 g sample was mixed in 3 mL of ethanol (90%), incubated at 70 °C and adjusted to a final volume of 25 mL with 90% ethanol. From this, 1 mL of the supernatant was mixed with 5 mL each of anthrone solution and sulfuric acid. Absorbance was measured at 485 nm, with glucose used as the standard [48].

Proline content was measured by mixing 0.1 g of the sample in 1 mL of sulfosalicylic acid solution. The reaction mixture included sample, glacial acetic acid, and ninhydrin each one with 0.25 mL, and final data were determined at 520 nm [49].

### 2.8. Statistical and Multivariate Analysis

A two-way ANOVA with a split-plot design was applied, where P and Cd were the main plot factors, and cotton genotypes were the subplot factors using Statistix 8.1 (Analytical Software, Tallahassee, FL, USA) software. Mean comparisons were performed using the least significant difference test at a 5% probability level. Graphs were generated with Graphpad Prism 8 (San Diego, CA, USA) and expressed as mean ± standard error based on three biological and technical replications. Principal component and correlation analysis were calculated in OriginPro (2015) (b9.2.214, OriginLab Corporation, Northampton, MA, USA).

## 3. Results

### 3.1. Plant Growth

Various growth traits were evaluated to assess the variation in morphological traits of cotton genotypes. Compared to the control, Cd stress significantly reduced shoot length, root dry weight, shoot dry weight, and total dry weight under low P condition. However, P supplementation alleviated the negative effects of Cd by improving growth parameters (Figure 1). Regardless of the treatment, Jimian169 consistently demonstrated greater shoot length, root dry weight, shoot dry weight, and total dry weight compared to DES926. Additionally, Cd stress negatively impacted root morphological traits, including root length, surface area, volume and diameter under both P conditions (Figure 2). The reductions were more pronounced in DES926, whereas Jimian169 maintained higher values for root morphological traits across both P conditions (Figure 2).

### 3.2. Leaf Chlorophyll and Gaseous Exchange

Cd stress significantly reduced leaf photosynthesis, transpiration, and stomatal conductance along with chlorophyll and carotenoid contents of contrasting low-P tolerant cotton genotypes under both P conditions, with the exception of intercellular CO_2_ concentration (Figure 3 and Figure 4). Regardless of treatments, Jimian169 exhibited superior photosynthetic performance compared to DES926 (Figure 3 and Figure 4). Under Cd stress, DES926 showed higher intercellular CO_2_ concentration, indicating reduced carboxylation efficiency despite the availability of CO_2_.

### 3.3. Phosphorus Concentration and Use Efficiency

Cd stress led to a significant reduction in root and shoot P concentration, particularly under low P conditions. However, Jimian169 maintained higher root and shoot P concentrations compared to DES926 (Figure 5A,B). A notable reduction in P accumulation of root and shoot was recorded in response to low and normal P relative to control. Jimian169 demonstrated greater root and shoot P accumulation than DES926 (Figure 5C,D). Furthermore, Cd stress reduced PUpE and PUtE, especially under low P conditions. Despite this, Jimian169 displayed significantly higher PUpE and PUtE than DES926 across both P conditions (Figure 5E,F).

### 3.4. Malondialdehyde Content and Reactive Oxygen Species

Cd stress markedly increased root and shoot malondialdehyde (MDA) content of cotton genotypes under low P. However, P supplementation significantly reduced MDA content, particularly in Jimian169 (Figure 6A,B). Regardless of the treatment, DES926 exhibited higher MDA content in roots and shoots compared to Jimian169 (Figure 6A,B). Similarly, Cd stress evaluated reactive oxygen species (ROS) levels, especially under low P conditions, whereas P supplementation effectively mitigated ROS accumulation, with Jimian169 showing the pronounced reduction. Compared to the control, Cd stress increased H_2_O_2_ concentration levels in roots and shoots under both low and normal P conditions with DES926 displaying higher H_2_O_2_ accumulation (Figure 6C,D). Similarly, Cd stress raised O_2_^−^ concentration levels in roots and shoots under both P conditions, with DES926 consistently exhibiting higher O_2_^−^ concentration levels than Jimain169 (Figure 6E,F).

### 3.5. Antioxidant Enzymatic Activities

Antioxidant enzymatic activities were measured to assess the tolerance of cotton genotypes against Cd stress across P concentration levels (Figure 7). Cd stress significantly suppressed the activities of SOD, POD, and CAT in both roots and shoots, particularly under low P conditions compared to control (Figure 7). However, P supplementation enhanced the activities of these enzymes, especially in Jimian169 (Figure 7). Comparative analysis revealed that Jimian169 exhibited higher SOD, POD, and CAT activities in roots and shoots than DES926 (Figure 7).

### 3.6. Osmo-Protectants

The levels of osmo-protectants were also evaluated. Cd stress significantly reduced free amino acids in roots and shoots under low P conditions. However, Jimian169 maintained significantly higher free amino acid concentration levels compared to DES926 (Figure 8A,B). Total soluble protein and total soluble sugar contents of roots and shoots were greatly improved under Cd stress across both P conditions. Regardless of treatment, Jimian169 showed higher concentration levels of total soluble protein and more total soluble sugars than DES926 (Figure 8C–F).

### 3.7. Multivariate Analysis

The principal component analysis (PCA) effectively discriminated among samples based on the two principal components, PC1 and PC2, highlighting the influence of Cd stress, P conditions, and cotton genotypes on plant responses. PC1 accounted for a significant portion of the total variation (80.63%) and primarily separated the samples based on external stress factors, particularly Cd exposure and P availability. Traits with high loading on PC1 included key growth parameters, leaf gaseous exchange, PUE, and antioxidant enzyme activities, indicating that these variables are major determinants of the plants’ response to Cd and P conditions. PC2, explaining 13.81% of the variation, provided additional discrimination among samples based on genotypic differences. Traits contributing strongly to PC2 included osmo-protectants and some antioxidant enzymes, suggesting that these characteristics are more genotype-dependent and play a secondary but notable role in stress adaptation. Overall, the PCA clearly separated treatments and genotypes, with PC1 representing environmental stress and nutrient factors, while PC2 reflected the inherent genetic variability response mechanism. The factor loading and scores together revealed that growth-related traits and physiological traits underlie the primary axis of variation, while biochemical stress markers define the secondary axis, offering insight into the differential tolerance mechanism among cotton genotypes (Figure 9 and Appendix A).

The correlation between plant growth, leaf gaseous exchange (excluding intercellular CO_2_ concentration), PUE traits, and antioxidant enzyme activities was strongly positive. Additionally, root and shoot growth, along with dry matter, showed strong positive correlations with most traits except for MDA, osmo-protectants, and ROS accumulation (Figure 10). Among root morphological traits, root length exhibited a strong positive correlation with PUE traits and antioxidant enzymes, highlighting the role of P in fostering an improved root system to enhance P uptake and tolerance to Cd stress (Figure 10).

## 4. Discussion

### 4.1. Phosphorus Enhances Morphophysiological Tolerance of Cotton Genotypes Under Cadmium Stress

Exposure to heavy or trace elements can lead to toxic effects on plant growth parameters, although the severity of these effects depends on plant species, genotype and tissue type. These growth attributes serve a sensitive indicators for assessing plant responses to metal toxicity [50]. Among heavy metals, cadmium (Cd) is a major pollutant in agricultural soils worldwide, significantly reducing plant growth and productivity [51]. The extent of Cd toxicity in plants varies based on growth conditions, experimental setup, Cd availability, exposure duration, and plant age [52].

In this study, Cd stress (5 µM) markedly reduced root and shoot length, plant height, and dry matter accumulation in two cotton genotypes. The results indicated that Jimian169 exhibited greater growth and dry matter accumulation than DES926 (Figure 1), suggesting a higher tolerance to Cd stress. In contrast, DES926 had low values for such parameters, indicating greater sensitivity, particularly under low P conditions. The observed growth reduction under Cd stress may be attributed to decreased chlorophyll contents, impaired photosynthesis, and related metabolic disruptions, ultimately leading to reduced carbohydrate production and biomass accumulation. These findings align with previous studies that reported significant declines in root and shoot length, as well as dry matter accumulation, under Cd stress [53,54]. Similar reductions were found in *Phyllanthus amarus* grown under high Cd stress [55].

A key early indicator of Cd toxicity in plants is visual symptomology, particularly leaf chlorosis, which results from structural damage to chloroplast [56]. It is generally accepted that Cd concentrations exceeding 5–10 mg/kg in leaf tissue are toxic to plants [57]. In this study, chlorosis was more evident than necrosis in both genotypes, with symptoms intensifying in a dose-dependent manner. DES926 exhibited more severe chlorosis than Jimian169, further confirming its higher sensitivity to Cd stress. However, some degree of natural chlorosis and necrosis were also observed in control plants. Based on visual observations, Jimian169 appeared more tolerant to Cd than DES926. Previous studies suggest that Cd-induced chlorosis may result from disruptions in essential nutrient accumulation in aerial plant parts [58].

The genotypic differences observed in growth responses and visual symptoms under Cd stress highlight distinct tolerance mechanisms in cotton genotypes. These findings are consistent with reports of Cd-induced chlorosis in various plant species, including chickpea [59], *Sedum alfredii* [60], rice [61], and mungbean [62]. The differential responses observed in this study provide valuable insights into morphological and physiological adaptations of cotton genotypes under Cd toxicity.

### 4.2. Phosphorus Enhances Antioxidant Defense System of Cotton Genotypes Under Cadmium Stress

Enzymatic activities showed different responses against Cd stress. The initial increase in antioxidant enzyme activity under Cd stress may be attributed to the activation of the plant defense mechanism in response to metal stress [63]. These antioxidant enzymes play a crucial role in mitigating oxidative damage by reducing MDA contents under metal stress [64]. However, at higher stress, the reduction in the antioxidant system suggests a disruption in ROS homeostasis, leading to elevated concentration levels of H_2_O_2_, MDA, and electrolyte leakage. Similar trends in antioxidant enzyme activity and ROS production have been observed in various plants under Cd stress [65,66].

Phosphorus (P) supplementation further enhanced antioxidant enzyme activities, consequently reducing oxidative stress as evidenced by lower MDA and H_2_O_2_ concentration levels in both roots and shoots under Cd stress (Figure 6). This P-induced enhancement of the antioxidant defense system has been reported under various abiotic stress conditions [65]. The observed inhibition of MDA and H_2_O_2_ accumulation with P application highlights its role in mitigating Cd-induced stress in cotton plants. Furthermore, the enhanced antioxidant enzyme activity in P-treated plants may be associated with reduced Cd in plants. These findings suggest that P-mediated stimulation of the antioxidant defense system and suppression of ROS production may constitute an essential tolerance mechanism against Cd-induced oxidative stress in cotton.

Under stress conditions, plants accumulate MDA, a marker of lipid peroxidation, which significantly damages cell membranes [67]. Thus, MDA serves as a reliable indicator of oxidative stress. In the current study, Cd stress led to elevated MDA concentration levels and antioxidant enzyme activities (Figure 6 and Figure 7). The increase in MDA content in Cd-stressed plants was likely due to excessive ROS accumulation. Previous studies have reported that abiotic stress disrupts ROS equilibrium, leading to enhanced photooxidative damage to the photosynthetic apparatus and membrane lipid peroxidation [68]. To maintain normal photosynthesis, plants must regulate ROS concentration levels [69] and maintain redox equilibrium in the cells [70]. One of the most critical mechanisms to achieve this balance is the antioxidant defense system [71].

In this study, cotton genotype Jimian169 exhibited a strong capacity to counter oxidative stress damage (Figure 6). Elevated antioxidant defense system and reduced ROS and MDA accumulation under normal P indicate that Jimian169 possesses a robust redox defense system against stress. Similarly, nitrogen application has been shown to enhance maize’s soluble protein and antioxidant enzyme activities under stress [72]. Thus, while Cd stress regulates antioxidant enzyme activity, its effectiveness largely depends on the presence of P.

Excessive ROS production and accumulation can lead to severe cellular damage resulting in cell death [7]. In this study, Cd-stressed plants exhibited significantly higher ROS concentration levels compared to control plants. Furthermore, combined low P and Cd stress exacerbated ROS and MDA accumulation, indicating that P application plays a crucial role in suppressing ROS production under Cd stress. The observed increase in ROS levels under Cd stress may be associated with reduced photosynthetic activity (Figure 3), consistent with previous findings [73]. To counteract oxidative stress, plants activate antioxidant defense mechanisms that help alleviate ROS-induced damage. Notably, Jimian169 exhibited higher antioxidant enzyme activities under Cd stress, suggesting a more effective antioxidant defense system compared to DES926 genotype. These findings indicate that DES926 has a weaker antioxidant response to Cd stress, whereas Jimian169 demonstrates a stronger capability to mitigate oxidative damage.

In response to Cd stress, plants accumulate various osmolytes to enhance osmotic potential, thereby maintaining cellular turgidity and physiological processes [74]. Cd stress also affects osmo-protectants, including free amino acids, total soluble proteins, and soluble sugars, which are critical for osmotic adjustment under stress conditions. Cotton seedlings modulate these osmo-protectants to maintain cellular osmoregulation, an essential trait for stress tolerance [49]. Changes in free amino acid levels in response to stress show its production, acting as a stress indicator [75]. Among these osmo-protectants, amino acids play a crucial role in sustaining cell turgidity and preventing dehydration [76]. The higher concentration levels of osmo-protectants in Jimian169 tissues contributed to osmotic adjustment, reducing osmotic potential and maintaining cell turgidity. These findings suggest that Jimian169 has a superior capacity to maintain free amino acid levels, thereby enhancing Cd stress tolerance.

Soluble proteins and sugars play a crucial role as cellular components that enhance plant resilience to stress [77]. Research has consistently shown that various plant species accumulate osmo-protectants when exposed to stressful conditions [36,78,79]. Consistent with these findings, our study revealed that Jimian169 seedlings subjected to Cd stress exhibited a significant increase in osmo-protectant production and accumulation. This response contributed to improved osmotic potential, sustained cell turgidity, and reduced tissue dehydration. As a result, Jimian169 demonstrates a more efficient osmo-protective mechanism, enhancing its tolerance to Cd stress.

## 5. Conclusions

In conclusion, Cd-induced stress had a profound impact on the growth and metabolic processes of contrasting low-P tolerant genotypes, as evidenced by reduced dry matter accumulation, photosynthetic activity, chlorophyll content, PUE, and antioxidant enzyme activity, along with an increase in MDA concentration levels. However, Jimian169 exhibited greater tolerance to Cd stress across both P conditions compared to DES926. Its enhanced growth, well-developed root system, higher photosynthetic efficiency, improved PUE, stronger antioxidant defense, and greater osmo-protectant accumulation contributed to its superior stress resilience. These findings highlight the critical role of P in alleviating Cd-induced toxicity and suggest that Jimian169 is a more suitable genotype for cultivation in Cd-contaminated soils with normal P.

## Figures and Tables

**Figure 1 antioxidants-14-00686-f001:**
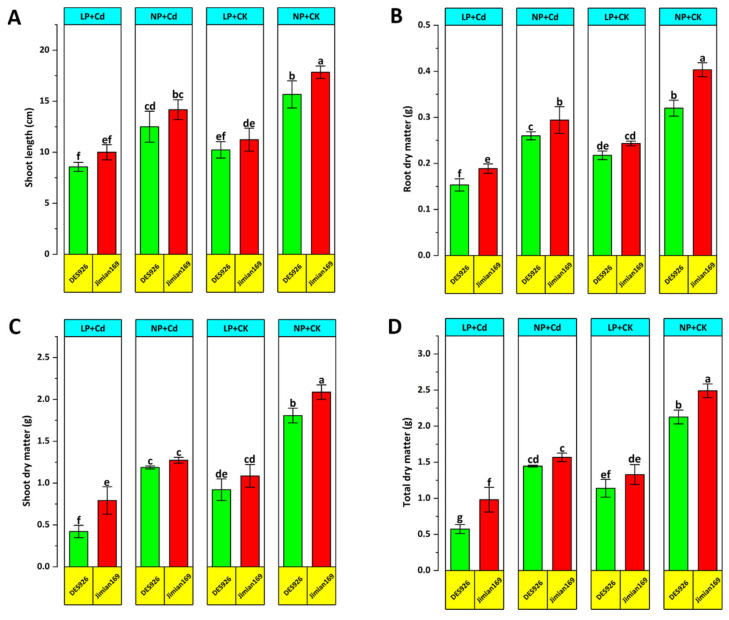
Effect of P supplementation on shoot length (**A**), root dry matter (**B**), shoot dry matter (**C**), and total dry matter (**D**) of Jimian169 and DES926 under Cd stress and control conditions. The values are presented as mean ± standard error. Error bars with different letters show significant differences (*p* < 0.05).

**Figure 2 antioxidants-14-00686-f002:**
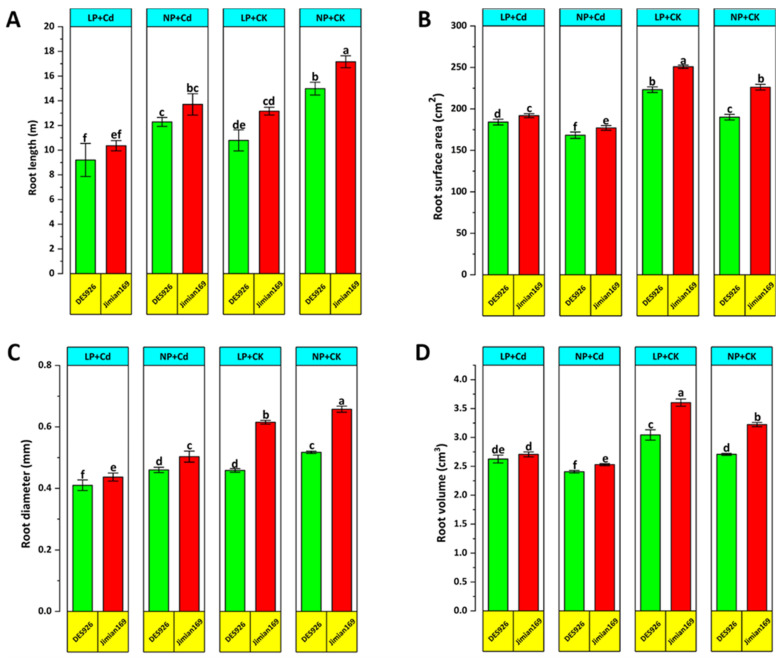
Effect of P supplementation on root length (**A**), root surface area (**B**), root diameter (**C**), and root volume (**D**) of Jimian169 and DES926 under Cd stress and control conditions. The values are presented as mean ± standard error. Error bars with different letters show significant differences (*p* < 0.05).

**Figure 3 antioxidants-14-00686-f003:**
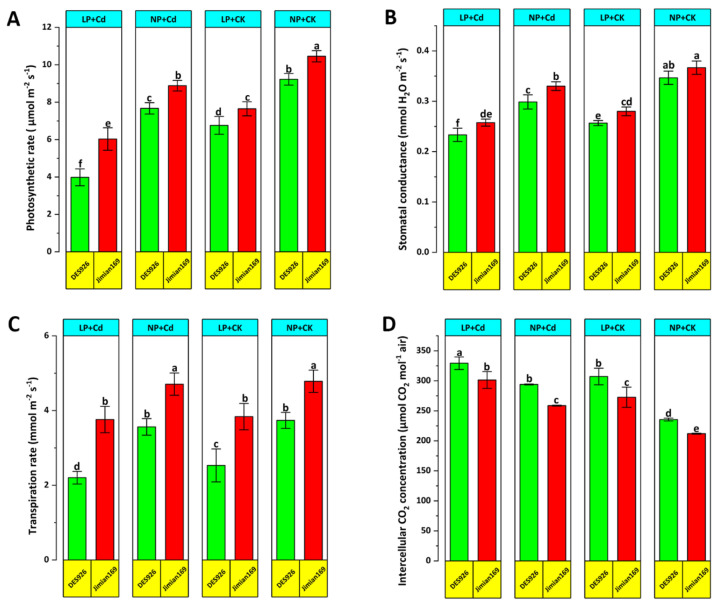
Effect of P supplementation on photosynthetic rate (**A**), stomatal conductance (**B**), transpiration rate (**C**), and intercellular CO_2_ concentration (**D**) of Jimian169 and DES926 under Cd stress and control conditions. The values are presented as mean ± standard error. Error bars with different letters show significant differences (*p* < 0.05).

**Figure 4 antioxidants-14-00686-f004:**
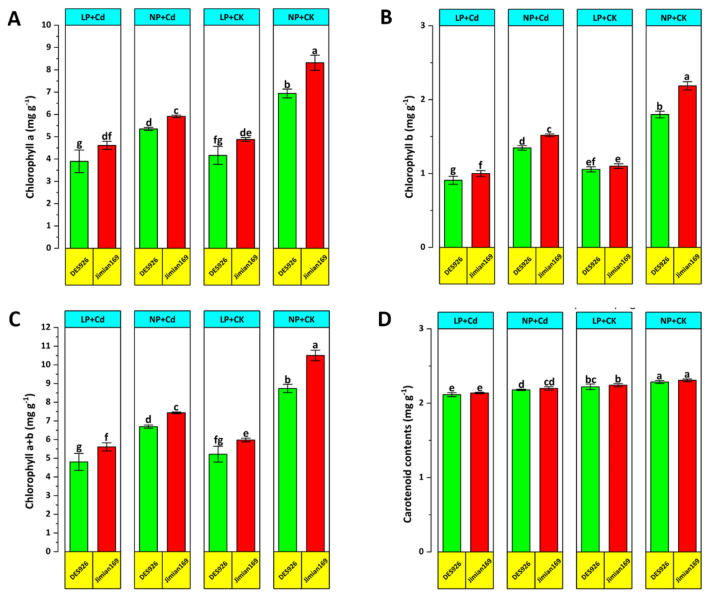
Effect of P supplementation on chlorophyll a (**A**), chlorophyll b (**B**), total chlorophyll (**C**), and carotenoid content (**D**) of Jimian169 and DES926 under Cd stress and control conditions. The values are presented as mean ± standard error. Error bars with different letters show significant differences (*p* < 0.05).

**Figure 5 antioxidants-14-00686-f005:**
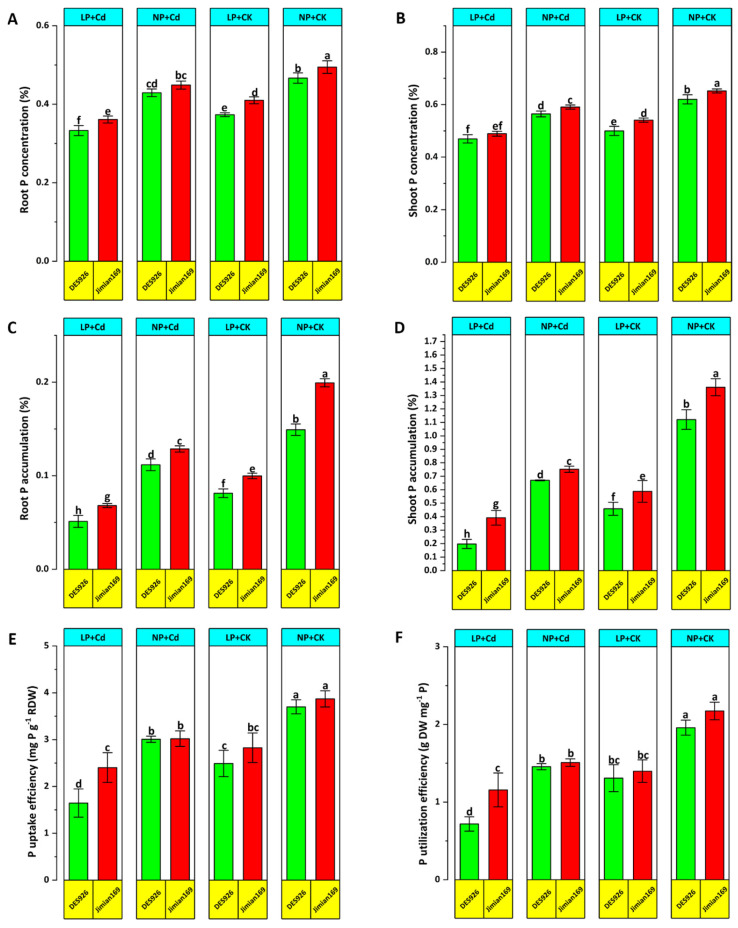
Effect of P supplementation on root P concentration (**A**), shoot P concentration (**B**), Root P accumulation (**C**), shoot P accumulation (**D**), P uptake efficiency (**E**), and P utilization efficiency (**F**) of Jimian169 and DES926 under Cd stress and control conditions. The values are presented as mean ± standard error. Error bars with different letters show significant differences (*p* < 0.05).

**Figure 6 antioxidants-14-00686-f006:**
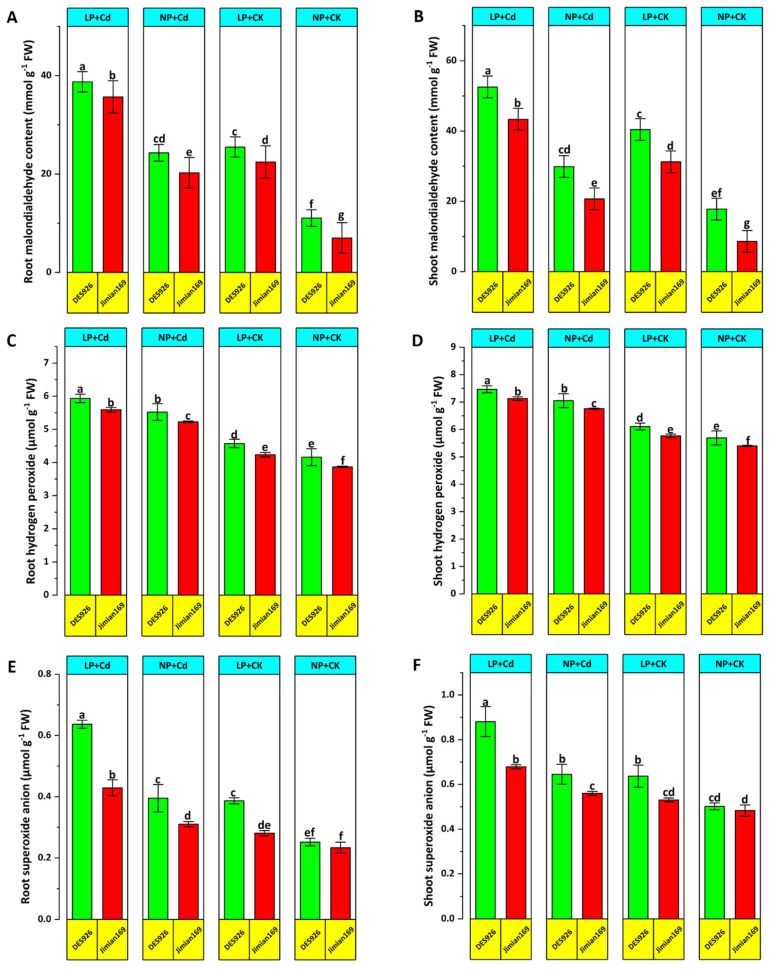
Effect of P supplementation on root MDA (**A**), shoot MDA (**B**), root H_2_O_2_ (**C**), shoot H_2_O_2_ (**D**), root O_2_^−^ (**E**), and shoot O_2_^−^ (**F**) of Jimian169 and DES926 under Cd stress and control conditions. The values are presented as mean ± standard error. Error bars with different letters show significant differences (*p* < 0.05).

**Figure 7 antioxidants-14-00686-f007:**
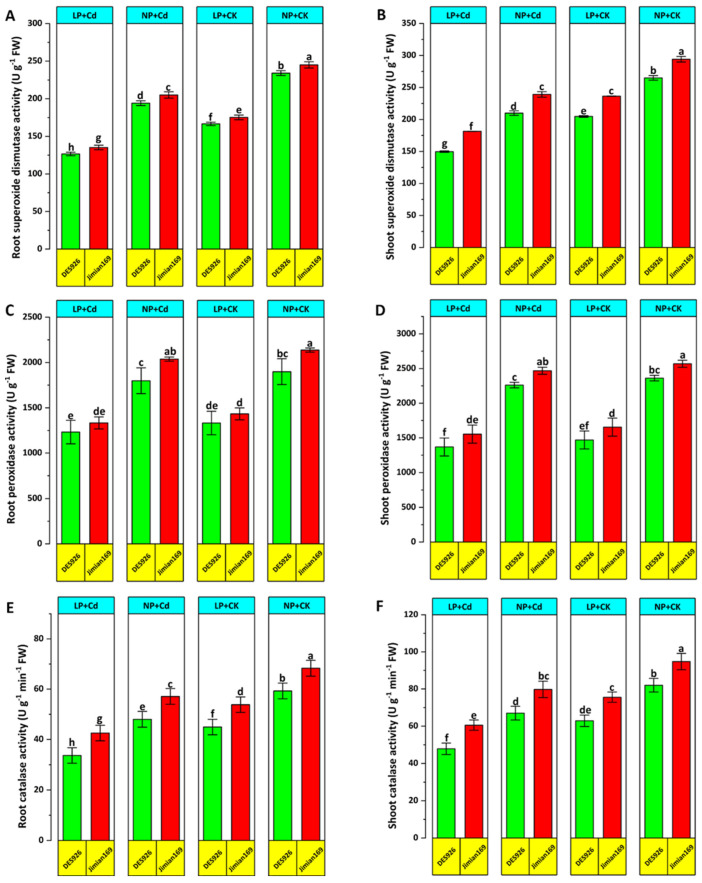
Effect of P supplementation on root SOD (**A**), shoot SOD (**B**), root POD (**C**), shoot POD (**D**), root CAT (**E**), and shoot CAT (**F**) of Jimian169 and DES926 under Cd stress and control conditions. The values are presented as mean ± standard error. Error bars with different letters show significant differences (*p* < 0.05).

**Figure 8 antioxidants-14-00686-f008:**
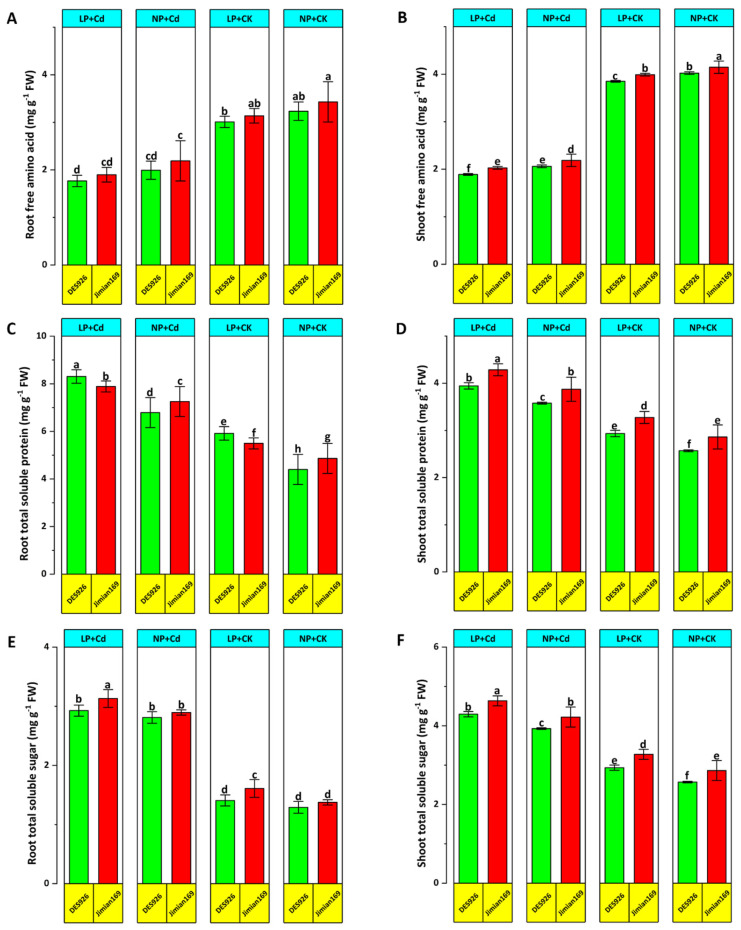
Effect of P supplementation on root-free amino acids (**A**), shoot-free amino acids (**B**), root total soluble protein (**C**), shoot total soluble protein (**D**), root total soluble sugar (**E**), and shoot total soluble sugars (**F**) of Jimain169 and DES926 under Cd stress and control conditions. The values are presented as mean ± standard error. Error bars with different letters show significant differences (*p* < 0.05).

**Figure 9 antioxidants-14-00686-f009:**
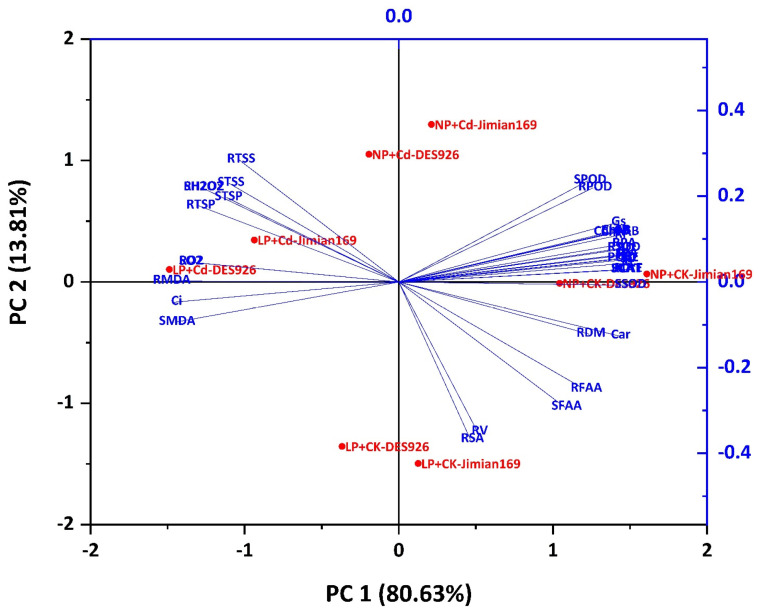
Principal component analysis (PCA) biplot of various studied traits of Jimian169 and DES926 under Cd stress and control supplemented with low and normal P conditions. SL; shoot length, RDM; root dry matter, SDM; shoot dry matter, TDM; total dry matter, RL; root length, RSA; root surface area, RD; root diameter, RV; root volume, Pn; photosynthetic rate, gs; stomatal conductance, E; transpiration rate, Ci; intercellular CO_2_ concentration, Chl a; chlorophyll a content, Chl b; chlorophyll b content, chlorophyll a + b, CAR; carotenoid contents, RP; root phosphorus concentrations, SP; shoot phosphorus concentration, RPA; root phosphorus accumulation, SPA; shoot phosphorus accumulation, PUpE; phosphorus uptake efficiency, PUtE; phosphorus utilization efficiency, RMDA; root malondialdehyde content, SMDA; shoot malondialdehyde content, RO_2_; root superoxide anion, SO_2_; shoot superoxide anion, RH_2_O_2_; root hydrogen peroxide, SH_2_O_2_; shoot hydrogen peroxide, RSOD; root superoxide dismutase, SSOD, shoot superoxide dismutase, RPOD; root peroxidase, SPOD; shoot peroxidase, RCAT; root catalase, SCAT; shoot catalase, RFAA; root free amino acids, SFAA; shoot free amino acids, RTSP; root total soluble proteins, STSP; shoot total soluble proteins, RTSS; root total soluble sugars, and STSS; shoot total soluble sugars.

**Figure 10 antioxidants-14-00686-f010:**
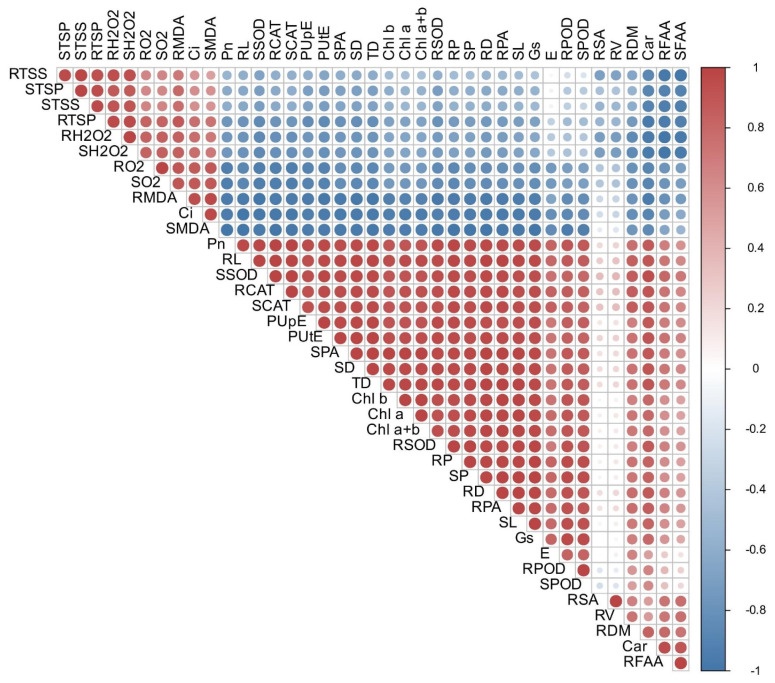
Relationships of various studied traits of Jimian169 and DES926 under Cd stress and control supplemented with low and normal P conditions. Where red color shows a positive correlation and blue color shows a negative correlation, while size of the circle shows the strength of correlation. SL; shoot length, RDM; root dry matter, SDM; shoot dry matter, TDM; total dry matter, RL; root length, RSA; root surface area, RD; root diameter, RV; root volume, Pn; photosynthetic rate, gs; stomatal conductance, E; transpiration rate, Ci; intercellular CO_2_ concentration, Chl a; chlorophyll a content, Chl b; chlorophyll b content, chlorophyll a + b, CAR; carotenoid contents, RP; root phosphorus concentrations, SP; shoot phosphorus concentration, RPA; root phosphorus accumulation, SPA; shoot phosphorus accumulation, PUpE; phosphorus uptake efficiency, PUtE; phosphorus utilization efficiency, RMDA; root malondialdehyde content, SMDA; shoot malondialdehyde content, RO_2_; root superoxide anion, SO_2_; shoot superoxide anion, RH_2_O_2_; root hydrogen peroxide, SH_2_O_2_; shoot hydrogen peroxide, RSOD; root superoxide dismutase, SSOD, shoot superoxide dismutase, RPOD; root peroxidase, SPOD; shoot peroxidase, RCAT; root catalase, SCAT; shoot catalase, RFAA; root free amino acids, SFAA; shoot free amino acids, RTSP; root total soluble proteins, STSP; shoot total soluble proteins, RTSS; root total soluble sugars, and STSS; shoot total soluble sugars.

## Data Availability

All data generated or analyzed during this study are included in this published article.

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
