# Peer review of "Phosphorus Supplementation Enhances Growth and Antioxidant Defense Against Cadmium Stress in Cotton"

_antioxidants, 2025, doi:10.3390/antiox14060686_

Round 1
Reviewer 1 Report
The authors address an important question in their publication: Can phosphorous (P) mitigate the toxicity of the heavy metal cadmium (Cd) in cotton? Cotton is a crop of industrial significance and Cd contamination is a serious environmental concern. This study therefore has great relevance. The authors chose two cotton genotypes, Jimian169 and DES926. Jimian169 is known to be tolerant towards low phosphorous (P) growing conditions, whereas DES926 is not. In addition to two cadmium concentration levels (control 0 mM and Cd stress at 5 mM) the authors also varied P-levels at 0.1 mM and 1 mM, respectively. The authors monitored growth, photosynthesis, P-use-efficiency, and antioxidant enzyme activities.
The introduction provided sufficient and interesting background information. In particular, I was intrigued by the idea that P could mitigate Cd toxicity while also enhancing its accumulation. It would have been interesting to check Cd levels in the plant tissues of the cotton. Maybe this idea will be pursued in a future study.
The experimental section was kept short by referring to procedures from previous studies by the authors or other research groups. In some cases, more than one study is cited per method which is a bit confusing. For example, lines 174-175: “Total soluble protein content was measured following the method of Bradford [49] with bovine serum albumin as the standard [50] and the absorbance was recorded at 595 nm [51].” Three references in one sentence for one commonly used method are too many. Please select the reference which most closely matches with your procedure and point out modifications if applicable. The same principal should be applied to the other paragraphs in the experimental section.
The results in sections 3.1 to 3.6 are presented in a very systematic and straightforward manner. Experiment by experiment the same type of graphical presentation is used consistently and the most important trends in the data are highlighted in the text. This makes it easy to follow along. The Principal Component Analysis and the Correlation plot in section 3.7 provide a good overview on the overall trends in the samples and the variables. However, what did not work at all for me was that the abbreviations used in Figures 9 and 10 are cumbersome to look up. Instead of listing the abbreviations within the supplement and lines 429 to 450, please move these abbreviations closer to these figures. Even if the figure captions would get really long, this would be better than having to turn to another location in the manuscript.
The discussion places the results of this study into the larger context of findings by other research teams and the conclusion summarizes the key results of this study in a compact and meaningful way.
Almost all of the references are all missing the journal name. The authors provide 87 references (which is a lot) and twelve of these references with Iqbal A. (first author here) as co-author. In searches for publications on similar topics. I came across a study that was no cited but has a lot of relevance:
AUTHOR=Li Ling , Yan Xuyu , Li Juan , Wu Xiang , Wang Xiukang
TITLE=Metabolome and transcriptome association analysis revealed key factors involved in melatonin mediated cadmium-stress tolerance in cotton
JOURNAL=Frontiers in Plant Science
VOLUME=Volume 13 - 2022
YEAR=2022
URL=https://www.frontiersin.org/journals/plant-science/articles/10.3389/fpls.2022.995205
Line 21 (abstract): Cd concentration given as 5 mM, but line 99 (introduction) mentions 5 µM Cd. Please resolve this discrepancy. Most likely 5 mM is the correct number as it also occurs in the Experimental section, however line 322 in the Discussion mentions 5 micromolar again.
Line 97: first occurrence of abbreviation PUE: Please define PUE here. I assume it means phosphorous use efficiency.
Line 133: Please provide citation or source for EPSON and WinRhizo.
Line 158: Different in-text citation style: Iqbal et al. (2023d) should be a number for consistency with other references.
Lines: 187 and 189: Please provide information on the software source for Statistix 8.1 and GraphPad Prism 8.
Line 374/5: Two different in-text citation styles “(Saidi et al., 2013;[75].” Please correct.
Author Response
Reviewer 1.
The authors address an important question in their publication: Can phosphorous (P) mitigate the toxicity of the heavy metal cadmium (Cd) in cotton? Cotton is a crop of industrial significance and Cd contamination is a serious environmental concern. This study therefore has great relevance. The authors chose two cotton genotypes, Jimian169 and DES926. Jimian169 is known to be tolerant towards low phosphorous (P) growing conditions, whereas DES926 is not. In addition to two cadmium concentration levels (control 0 mM and Cd stress at 5 mM) the authors also varied P-levels at 0.1 mM and 1 mM, respectively. The authors monitored growth, photosynthesis, P-use-efficiency, and antioxidant enzyme activities.
Author response to reviewer 1:
We would like to thank the reviewer for his deep reading and useful comments, which helped us to strongly improve the clarity of the article. We have revised the manuscript accordingly and point-by-point responses are as follows:
The introduction provided sufficient and interesting background information. In particular, I was intrigued by the idea that P could mitigate Cd toxicity while also enhancing its accumulation. It would have been interesting to check Cd levels in the plant tissues of the cotton. Maybe this idea will be pursued in a future study.
Response: We appreciate the reviewer’s insightful comment and interest in the potential role of phosphorus in mitigating cadmium toxicity while influencing its accumulation. We agree that measuring Cd concentrations in plant tissues would have provided valuable insights. We acknowledge its importance and intend to incorporate this aspect into our future research. Thank you for the helpful suggestion.
The experimental section was kept short by referring to procedures from previous studies by the authors or other research groups. In some cases, more than one study is cited per method which is a bit confusing. For example, lines 174-175: “Total soluble protein content was measured following the method of Bradford [49] with bovine serum albumin as the standard [50] and the absorbance was recorded at 595 nm [51].” Three references in one sentence for one commonly used method are too many. Please select the reference which most closely matches with your procedure and point out modifications if applicable. The same principal should be applied to the other paragraphs in the experimental section.
Response: We appreciate the reviewer’s constructive feedback. We agree that citing multiple references for a single standard method may cause confusion. Accordingly, we have revised the experimental section to retain only the most relevant reference for each method and have specified any modifications where applicable. Thank you.
The results in sections 3.1 to 3.6 are presented in a very systematic and straightforward manner. Experiment by experiment the same type of graphical presentation is used consistently and the most important trends in the data are highlighted in the text. This makes it easy to follow along. The Principal Component Analysis and the Correlation plot in section 3.7 provide a good overview on the overall trends in the samples and the variables. However, what did not work at all for me was that the abbreviations used in Figures 9 and 10 are cumbersome to look up. Instead of listing the abbreviations within the supplement and lines 429 to 450, please move these abbreviations closer to these figures. Even if the figure captions would get really long, this would be better than having to turn to another location in the manuscript.
Response: We thank the reviewer for the positive feedback on the clarity and consistency of our results presentation. We also appreciate the helpful suggestion regarding the abbreviations in Figures 9 and 10. In response, we have moved all relevant abbreviations directly into the figure captions to improve readability and eliminate the need to refer to other sections or supplementary material.
The discussion places the results of this study into the larger context of findings by other research teams and the conclusion summarizes the key results of this study in a compact and meaningful way.
Response: We sincerely thank the reviewer for the positive remarks and appreciation of our discussion and conclusion. We are glad that the context and clarity of our findings were well received.
Almost all of the references are all missing the journal name. The authors provide 87 references (which is a lot) and twelve of these references with Iqbal A. (first author here) as co-author. In searches for publications on similar topics. I came across a study that was no cited but has a lot of relevance:
AUTHOR=Li Ling , Yan Xuyu , Li Juan , Wu Xiang , Wang Xiukang
TITLE=Metabolome and transcriptome association analysis revealed key factors involved in melatonin mediated cadmium-stress tolerance in cotton
JOURNAL=Frontiers in Plant Science
VOLUME=Volume 13 - 2022
YEAR=2022
URL=https://www.frontiersin.org/journals/plant-science/articles/10.3389/fpls.2022.995205
Response: We thank the reviewer for pointing this out. We have corrected the formatting of the references to include journal names where missing. Additionally, we have reviewed and streamlined the reference list to remove redundancies and have now included the suggested relevant study by Li et al. (2022) in the revised manuscript. We appreciate the helpful recommendation.
Detailed comments
Line 21 (abstract): Cd concentration given as 5 mM, but line 99 (introduction) mentions 5 µM Cd. Please resolve this discrepancy. Most likely 5 mM is the correct number as it also occurs in the Experimental section, however line 322 in the Discussion mentions 5 micromolar again.
Response: We sincerely apologize for the typographical error. The correct concentration used in the study is 5 µM Cd, and we have now corrected this inconsistency throughout the manuscript to ensure accuracy. Thank you for bringing this to our attention.
Line 97: first occurrence of abbreviation PUE: Please define PUE here. I assume it means phosphorous use efficiency.
Response: We thank the reviewer for pointing this out. The abbreviation PUE has now been defined as phosphorus use efficiency at its first occurrence in the manuscript.
Line 133: Please provide citation or source for EPSON and WinRhizo.
Response: We have provided the reference for the above machine and software used in this study. Thank you.
Line 158: Different in-text citation style: Iqbal et al. (2023d) should be a number for consistency with other references.
Response: Thank you for the observation. We have now included appropriate references and source information for the EPSON scanner and WinRhizo software used in this study.
Lines: 187 and 189: Please provide information on the software source for Statistix 8.1 and GraphPad Prism 8.
Response: Thank you for your helpful comment. We have now added the full source details for Statistix 8.1 and GraphPad Prism 8 in the revised manuscript, including the developer names and locations.
Line 374/5: Two different in-text citation styles “(Saidi et al., 2013;[75].” Please correct.
Response: Thank you for pointing out the inconsistency. We have corrected the citation to ensure a consistent referencing style throughout the manuscript.

Reviewer 2 Report
This study aimed at evaluating the effects of phosphorus (P) supplementation on the growth, photosynthesis, P use efficiency (PUE), and antioxidant enzyme activities of 2 cotton genotypes (Jimian169 and DES926) under cadmium (Cd) stress. A hydroponic experiment was performed at 2 levels of Cd concentration (0 and 5 μM), with and without P supplementation.
The study is interesting, the subject is topical in the related field, the methodology is adequate, there is no redundant information, the results are significant, the conclusions are supported by the data presented, but the text should be carefully revised and some changes should be made. I suggested some modifications in the attached document.
The main modifications are as follows:
- Table S1 should be added in the manuscript.
- PCA analysis should be detailed, i.e., discrimination on PC1 and PC2 between samples and their levels of relevant variables (characteristics) should be highlighted depending on the factor loadings and scores.
- It should be specified what the lowercase letters in Figures 1-8 mean.
- The symbols of variables in Figures 9 and 10 should be explained.
- Concentration level should be used instead of level (level means value) in the text as well as in the captions and on Oy axis of the Figures 1-8.
See the attached file.

Author Response
Reviewer 2 comments
This study aimed at evaluating the effects of phosphorus (P) supplementation on the growth, photosynthesis, P use efficiency (PUE), and antioxidant enzyme activities of 2 cotton genotypes (Jimian169 and DES926) under cadmium (Cd) stress. A hydroponic experiment was performed at 2 levels of Cd concentration (0 and 5 μM), with and without P supplementation.
The study is interesting, the subject is topical in the related field, the methodology is adequate, there is no redundant information, the results are significant, the conclusions are supported by the data presented, but the text should be carefully revised and some changes should be made. I suggested some modifications in the attached document.
Author response to reviewer 2:
We would like to thank the reviewer for his deep reading and useful comments, which helped us to strongly improve the clarity of the article. We have revised the manuscript accordingly and point-by-point responses are as follows:
The main modifications are as follows:
Table S1 should be added in the manuscript.
Response: Thank you for the suggestion. We have added Table S1 as a supplementary file to provide detailed PCA analysis data while avoiding redundancy in the main manuscript.
PCA analysis should be detailed, i.e., discrimination on PC1 and PC2 between samples and their levels of relevant variables (characteristics) should be highlighted depending on the factor loadings and scores.
Response: We thank the reviewer for this valuable suggestion. We have expanded the PCA analysis section to include detailed interpretation of the discrimination between samples along PC1 and PC2, emphasizing the relevant variables based on factor loadings and scores as recommended.
It should be specified what the lowercase letters in Figures 1-8 mean.
Response: Thank you for the helpful comment. We have now added an explanation of the lowercase letters to the captions of Figures 1–8 to clarify their meaning.
The symbols of variables in Figures 9 and 10 should be explained.
Response: Thank you for your comment. We have now defined all variable symbols and abbreviations directly in the captions of Figures 9 and 10 for clarity.
Concentration level should be used instead of level (level means value) in the text as well as in the captions and on Oy axis of the Figures 1-8.
Response: We thank the reviewer for this helpful suggestion. The term “concentration level” has been added throughout the text to improve clarity.

Round 2
Reviewer 1 Report
All my previous concerns have been addressed.
No more minor mistakes were detected
Reviewer 2 Report
The paper was improved and I think it can be published in this form.
The paper was improved and I think it can be published in this form.